# Peanut and Peanut-Based Foods Contamination by Toxigenic Fungi and Mycotoxins: Potential Risks for Beninese Consumers

**DOI:** 10.3390/toxins17110532

**Published:** 2025-10-29

**Authors:** Christin Sogbossi Gbétokpanou, Camille Jonard, Ornella Anaïs Mehinto, Sébastien Gofflot, Mawougnon Jaurès Martial Adjéniya, Ogouyôm Herbert Iko Afe, Dona Gildas Anihouvi, Samiha Boutaleb, Claude Bragard, Paulin Azokpota, Jacques Mahillon, Marianne Sindic, Marie-Louise Scippo, Yann Eméric Madodé, Caroline Douny

**Affiliations:** 1Laboratory of Food Analysis, Fundamental and Applied Research for Animals and Health (FARAH), Faculty of Veterinary Medicine, University of Liège, 4000 Liège, Belgium; sopissi2@yahoo.fr (C.S.G.); sboutaleb@uliege.be (S.B.); cdouny@uliege.be (C.D.); 2Laboratory of Food Sciences and Technologies, Ecole Doctorale des Sciences Agronomiques et de l’Eau, University of Abomey-Calavi, Abomey-Calavi 03 BP 2819, Benin; ornella.mehento@gmail.com (O.A.M.); martial.adjeniya@gmail.com (M.J.M.A.); ikohsoft@yahoo.fr (O.H.I.A.); donagildas@gmail.com (D.G.A.); azokpotap@yahoo.fr (P.A.); yann.madode@gmail.com (Y.E.M.); 3Laboratory of Agro-Food Quality and Safety, Axis Chemistry for Sustainable Food and Environmental Systems, Gembloux Agro-Bio Tech, Université de Liège, Passage des Déportés 2, 5030 Gembloux, Belgium; marianne.sindic@uliege.be; 4Knowledge and Valorization of Agricultural Products Department, Walloon Agricultural Research Centre, 5030 Gembloux, Belgium; c.jonard@cra.wallonie.be (C.J.); s.gofflot@cra.wallonie.be (S.G.); 5Laboratory of Plant Health-Applied Microbiology, Earth and Life Institute, Université Catholique de Louvain, 1348 Louvain-la-Neuve, Belgium; claude.bragard@uclouvain.be; 6Laboratory of Food and Environmental Microbiology, Earth and Life Institute-Applied Microbiology, Université Catholique de Louvain, 1348 Louvain-la-Neuve, Belgium; jacques.mahillon@uclouvain.be

**Keywords:** *Arachis hypogaea*, AFB1, total aflatoxins, OTA, MOE, risk assessment, process contribution, *kluiklui*

## Abstract

This study assessed mycotoxin contamination in roasted peanut snacks and *kluiklui* (fried pressed peanut cake), and consumer exposure in southern Benin. Roasted peanut snacks and *kluiklui* were sampled from markets across six municipalities, and their production follow-up was conducted on two sites using different processing methods. Mycotoxins were quantified using UPLC-MS/MS, while fungal species were identified via culture-based methods. Exposure to aflatoxin B1, total aflatoxins and ochratoxin A was estimated. Aflatoxin B1 predominated, reaching 169 µg/kg in roasted peanut snacks and 2144.64 µg/kg in marketed *kluiklui.* In contrast, just-produced *kluiklui* contained much lower levels (11.73–37.78 µg/kg). *Aspergillus flavus* and *Aspergillus niger* predominated in *kluiklui* from the first processing site, while *Aspergillus chevalieri* dominated in *kluiklui* from the second processing site. The grinding step (using public grinder) was identified as the main contamination point. The significative higher mycotoxin levels in *kluiklui* sampled on markets compared to just-produced *kluiklui* are probably due to poor storage conditions. Dietary exposure estimates revealed that margins of exposure for aflatoxins were far below the safety threshold of 10,000, and liver cancer risk estimates were particularly high for *kluiklui* consumers. *Kluiklui* consumption poses a significant health risk in Benin. Improved hygiene in public grinders and better storage practices are urgently needed to reduce contamination and protect consumers’ health.

## 1. Introduction

Mycotoxin contamination of food crops represents one of the most significant threats to global food safety and human health [1]. Mycotoxins are toxic secondary metabolites mainly produced by fungal species of the genera *Aspergillus* spp., *Fusarium* spp., and *Penicillium* spp., and to a lesser extent by *Alternaria* spp., *Claviceps* spp., *Trichoderma* spp. and *Trichothecium* spp. [2,3,4,5]. A single fungal species can produce multiple mycotoxins, potentially leading to the simultaneous contamination of food commodities with several toxins. In addition, successive colonisation by different fungal species can further contribute to multi-mycotoxin contamination [2]. To date, over 400 mycotoxins have been identified, but only a limited number are considered of major concern for food safety due to their documented occurrence in food products. These include aflatoxins B1 (AFB1), B2 (AFB2), G1 (AFG1), G2 (AFG2); ochratoxins, mainly ochratoxin A (OTA) and fumonisins, notably B1 (FB1) and B2 (FB2) [6,7].

Aflatoxins are mainly produced by *Aspergillus flavus* and *Aspergillus parasiticus*. Aflatoxins are classified by the International Agency for Research on Cancer (IARC) as carcinogenic to humans (Group 1) and present hepatotoxic and immunosuppressive properties [8]. Aflatoxin B1 is the most frequently present in contaminated foods and is recognised as the most toxic and carcinogenic. Its metabolite (AFB1 8,9-epoxide) displays several biological activities, including acute toxicity, teratogenicity, mutagenicity and carcinogenicity [development of hepatocellular carcinoma (HCC)] in humans and animals [9]. AFB1 has also been associated with growth suppression and weight loss in children [10] and immune system modulation [11]. The risk of development of HCC due to aflatoxin B1 exposure is higher for people infected with hepatitis B virus (HBV) who present a higher risk of developing liver cancer [12]. Ochratoxins (OTs) are produced by *Aspergillus ochraceus* and some *Penicillium species*, with OTA being the most extensively studied. OTA primarily affects the kidneys and is classified as a possible human carcinogen by the IARC (group 2B) [13]. Fumonisins, primarily produced by *Fusarium verticillioides* and *Fusarium proliferatum*, are also classified in the group 2B of possible human carcinogens [14]. Among them, fumonisin B1 (FB1) is the most prevalent and has been associated with both nephrotoxicity and hepatotoxicity in various animal species [15,16].

The main route of mycotoxin exposure for humans is through ingestion of contaminated food [17]. Fungal contamination and mycotoxin production can occur at any stage of the food production chain, including pre-harvest, harvest, drying, and storage phases [18]. Crops such as maize, millet, wheat, sorghum, soybean, peanut, and their derived products are among the most susceptible to mycotoxin contamination [19].

Peanuts (*Arachis hypogaea*) are one of the most widely grown oilseed crops in the world, cultivated across the intertropical zone [20]. Peanuts are important in terms of nutritional and economic aspects, particularly in developing countries [21] where they are processed into a wide range of popular food products [22,23]. Peanuts are consumed fresh, boiled, roasted, or processed into paste, oil or fried pressed-peanut cakes [22,23,24]. Besides peanut economic and nutritional benefits, peanut and peanut-based foods consumption has potential drawbacks due to their high susceptibility to fungal contamination, particularly by *A. flavus*, which produces aflatoxins [25,26].

Weather conditions are favourable to mycotoxin development in peanuts in Africa [27]. Studies across West Africa (Senegal, Ghana, Mali, Nigeria, Togo, and Benin) have reported mycotoxin contamination in peanuts and peanut-based foods with mycotoxins [28,29,30,31,32,33,34,35]. The reported concentrations widely exceeded the maximum limits established by the Commission Regulation (EU) 2023/915, which are set at 8 µg/kg for aflatoxin B_1_ and 15 µg/kg for total aflatoxins in raw peanuts intended for direct human consumption, and 2 µg/kg and 4 µg/kg, respectively, in peanut-based processed products [36]. Regular consumption of peanut-based foods may lead to a public health risk.

Also, in West Africa, particularly in Benin, peanut-based foods are primarily obtained through traditional, small-scale processing methods [37]. Mycotoxin levels in peanut-based foods may therefore stem from several factors, including the poor quality of peanut seeds used as raw material, inadequate technological processing practices, and/or inappropriate storage conditions of peanut-based foods.

Although a few studies in Benin have reported the presence of mycotoxins in peanut-based foods such as *kluiklui* (fried pressed peanut cake) and peanut oil [28,32], there is a significant lack of research assessing the contribution of processing practices to the contamination of these derived products. Moreover, data remain scarce regarding the dietary exposure of Beninese consumers to mycotoxins through peanut-based food consumption. This study, therefore, aims to address these gaps by evaluating the occurrence of mycotoxins and fungal contamination in peanuts and their derived products, and by assessing consumer exposure resulting from their consumption.

## 2. Results and Discussion

### 2.1. Occurrence of Mycotoxins in Marketed Roasted Peanut Snack and Kluiklui

Figure 1 illustrates the concentration ranges of main mycotoxins quantified in market samples of roasted peanut snacks and *kluiklui* in Benin, i.e., AFB1, AFB2, AFG1, AFG2, and OTA and OTB. The detailed results for all analysed mycotoxins are presented in Appendix A.

AFB1, AFB2, and AFG1 were the most frequently detected mycotoxins in roasted peanut snacks, with respective concentrations ranging from 0.02 to 169.0 µg/kg, 0.07 to 35.3 µg/kg, and 0.16 to 1.7 µg/kg, respectively, and corresponding median concentrations of 1.0, 0.07, and 0.16 µg/kg, respectively. AOH, FUM B1 and B2 were never detected in roasted peanut (Figure 1; Appendix A). One particular sample (ID: 20230.0S264, from Aplahoué) displayed inexplicably very high concentrations of OTA and OTB (5085 and 2143 µg/kg, respectively) (Figure 1; Appendix A.

In *kluiklui* samples, AFB1, AFB2, AFG1, AFG2, OTA and OTB were detected in almost all samples (Appendix A). Their concentration ranges were 89.1–2144.6 µg/kg for AFB1, 0.07–416.4 µg/kg for AFB2, 0.16–76.8 µg/kg for AFG1, 0.13–13 µg/kg for AFG2, 0.4–668 µg/kg for OTA, and 0.4–909 µg/kg for OTB (Appendix A). The corresponding median upper bound (UB) concentrations were 394.8, 83.9, 12.8, 1.7, 23.3, and 16 µg/kg, respectively. The minimum, median, and maximum total aflatoxin UB concentrations were 0.38, 1.36 and 169.36 µg/kg for roasted peanuts, while they were 112.1, 478.4 and 2644.1 µg/kg for *kluiklui* (Appendix A.

Several studies have identified AFB1, AFB2, AFG1, and AFG2 as the main mycotoxins frequently occurring in *kluiklui*, often in conjunction with ochratoxin A (OTA) [28,38]. According to Adjou et al. [28], the concentrations of AFB1, AFB2, AFG1, AFG2, and OTA in *kluiklui* marketed in Benin in 2011 reached 455, 491, 100, 88 and 2 µg/kg, respectively. Njumbe Ediage et al. [38] reported maximum concentrations of 282 µg/kg for AFB1, 31 µg/kg for AFB2, 79 µg/kg for AFG1, 96 µg/kg for AFG2, and 2 µg/kg for OTA in *kluiklui* samples collected in Benin in 2010. In Nigeria, a neighbouring country of Benin, in 2011, AFB1 and AFG2 concentrations of 2820 and 477 µg/kg, respectively, were reported in *kluiklui* [31].

AFB1, AFB2, AFG1 and AFG2 were quantified in roasted peanuts sold in Nigeria, with concentrations reaching up to 165, 26, 20 and 10 µg/kg, respectively [30]. In Iran, concentrations of 244, 6.40, 69.25 and 3.45 µg/kg of AFB1, AFB2, AFG1 and AFG2, respectively, have been reported in marketed roasted peanuts in 2015/2016 [39]. Sombie, et al. [40] also reported the presence of AFB1, AFB2, AFG1 and AFG2 in marketed roasted peanut samples, with concentrations reaching up to 1387, 271, 3328 and 742 µg/kg, respectively, in Serra Leone in 2017. None of these studies reported the presence of ochratoxins in roasted peanut snacks, in contrast to the findings of this study. Benin, Nigeria, and Sierra Leone are West African countries with a tropical climate, under the influence of the Atlantic Ocean, and showing regional variations [41]. These countries are very vulnerable to climate change, which favours mycotoxin contamination in peanuts and peanut-based foods, causing a possible increase in cancer incidence [42,43].

In Europe, no maximal limit exists for ochratoxins in peanut-based food, while for aflatoxins, the European Commission (EC) set a maximum concentration of 2 µg/kg for AFB1 and 4 µg/kg for total aflatoxins (i.e., the sum of AFB1, B2, G1 and G2) in peanut-based food for the final consumer [36]. Among samples above LOQ, all *kluiklui* samples and 41% of roasted peanut snacks samples had AFB1 concentrations exceeding the established regulatory limits (Figure 1A), with maximum levels of AFB1 being 80 and 1000 times higher than the limit, respectively.

### 2.2. Contribution of Processing Practices to Mycotoxin Contamination

#### 2.2.1. Occurrence of Mycotoxins in Raw Peanuts Used for Kluiklui and Roasted Peanut Snacks Production

Table 1 presents mycotoxin concentrations of raw peanuts used for *kluiklui* and roasted peanut snacks production. Since preliminary analyses revealed mycotoxin contamination of pools 11 and 12, individual samples forming these pools were analysed. Thus, pools 1 to 10, the three individual samples forming pool 11, and the three samples forming pool 12 were analysed, for a total of 16 raw material samples (10 pools and 6 individual samples). Appendix A presents the individual data for the raw material samples analysed. Concentrations of AFB1 and total aflatoxins (AFtot) ranged from 0.02 (LOD) to 2.63 µg/kg, and from 0.09 to 2.97 µg/kg, respectively. In case of raw peanuts (to be sorted before human consumption), the EC has established maximum levels of 8 µg/kg for aflatoxin B1 and 15 µg/kg for total aflatoxins [36]. Similarly, in Benin, Decree 2007 No. 0362 MAEP/D-CAB/SGM/DRH/DP/SA, which sets the maximum levels for contaminants in foodstuffs, has also established the same limits for aflatoxin B1 (8 µg/kg) and total aflatoxins (15 µg/kg) [44]. Aflatoxin concentrations in raw peanuts used to produce *kluiklui* and roasted peanut snacks in this study are below these EC and Benin Government limits. The concentrations of AOH, FUM B1, FUM B2, OTA, and OTB were below the limits of quantification for all analysed raw material samples.

#### 2.2.2. Kluiklui Processing Practices and Mycotoxins Contamination

The mycotoxin contamination of *kluiklui* samples collected during processing practices is presented in Table 2 and Appendix A. All roasted peanut samples (roasting being the first step of the process, see Figure 4) were below the LOD for all the mycotoxins analysed, showing that roasting did not contribute to mycotoxin contamination during *kluiklui* production process.

On the contrary, aflatoxins were quantified in all peanut paste samples obtained after dehulling and grinding, which are the next steps of the process (Figure 4). Ochratoxins were found as well, but not in all samples, while fumonisins (B1 and B2) and AOH were below the LOD in all peanut paste samples (Table 2 and Appendix A). The mean concentrations of AFB1, total aflatoxins, and OTA were 50.56, 60.91 and 3.70 µg/kg, respectively, for peanut paste from the processing without maize flour; and 14.65, 21.60 and 6.13 µg/kg, respectively, for peanut paste from the processing with maize flour. These findings suggest a contamination occurring during the grinding of roasted peanut seeds, as reported by Ndung’u, et al. [45]. Indeed, the grinding is carried out using a public grinder, originally intended for maize or spice grinding. The use of such shared equipment may lead to cross-contamination, as various raw materials of differing hygienic quality are processed in the same grinder. Additionally, as these are public facilities, processors have little control over the sanitary conditions and maintenance of the grinding equipment.

Aflatoxin B1 and total aflatoxin concentrations were significantly higher in peanut paste samples collected from the process without maize flour; with respective *p*-values of 0.04718 and 0.0479 for AFB1 and total aflatoxins. During peanut processing without maize flour, conducted in the municipality of Covè, the public grinders used were exclusively dedicated to *kluiklui* production. In contrast, in Aplahoué, the process involving maize flour relied on mixed-use grinders, which were not reserved exclusively for peanuts but were also used for maize grinding. As these grinders in the municipality of Aplahoué are shared with maize processing, a cleaning step is systematically performed prior to peanut grinding, to remove residual maize flour. However, in the municipality of Covè, since the grinders are used exclusively for peanut processing, thorough cleaning is not routinely performed prior to grinding. This could explain the higher level of contamination observed in peanut paste from the process without maize flour. Moreover, the warm and humid climate in southern Benin favours the increase in contamination with fungi and mycotoxins in peanut and ground maize residues from public grinders.

Regarding the maize flour used in *kluiklui* production, all collected samples were contaminated with aflatoxins and fumonisins, and showed important variation among samples (Table 2 and Appendix A). The mean concentrations were 24.11 µg/kg for AFB1, 27.26 µg/kg for AF tot, 991.57 µg/kg for FUM B1, and 333.84 µg/kg for FUM B2. Contamination of maize by fumonisins has also been reported in other studies conducted in Benin, with concentrations of fumonisin B1 and B2 reaching up to 836 and 221 µg/kg, respectively [38], and as high as 14,200 mg/kg and 3750 mg/kg, respectively, in earlier findings [46].

Regarding the final product of both processing methods, all *kluiklui* samples were contaminated with AFB1 and AFB2. OTA and OTB were found in a total of 12 samples out of 18, FUMB1 and B2 in 3 and 2 samples out of the 9 processed with the addition of maize flour, while AOH was not found in any sample. *Kluiklui* produced without maize flour and with maize flour had mean AFB1 concentrations of 37.78 and 11.73 µg/kg, respectively; and mean total aflatoxin concentrations of 46.99 and 16.24 µg/kg, respectively. AFB1 (37.78 vs. 11.73 µg/kg) and total aflatoxins concentration (46.99 vs 16.24 µg/kg) were significantly higher in *kluiklui* samples produced without the addition of maize flour, compared to those produced with maize flour (Table 2). The corresponding *p*-values were 0.04787 and 0.04584, respectively, for AFB1 and AFtot. These results are in line with those reported by Kayode, et al. [47], who observed higher average concentrations of total aflatoxins in peanut-based snacks (620.9 µg/kg) than in peanut/maize-based snacks (12 µg/kg).

The low levels of FUM B1 and FUM B2 found in some samples of *kluiklui* produced using maize flour could be explained by the low amount of maize flour added to the peanut paste (about 10 g of maize flour per kg of peanut paste).

AFB1, AFtot and OTA concentrations were significantly lower in just-produced *kluiklui* (from production follow-up) than marketed *kluiklui* with *p*-values of 2.158 × 10^−6^, 1.924 × 10^−6^ and 0.001096 for AFB1, AFtot and OTA, respectively (Appendix A). These findings suggest that inadequate post-production storage conditions may play a significant role in the contamination of *kluiklui* with mycotoxins.

#### 2.2.3. Roasted Peanut Processing Practices and Mycotoxin Contamination

Table 3 and Appendix A present the contribution of roasting practices (with or without heat transfer material) to mycotoxin contamination in roasted peanut snacks.

AFB1 and AFtot concentrations were 0.50 and 1.56 µg/kg, respectively, in roasted peanut snack samples processed without a heat transfer material; whereas concentrations of 0.12 and 0.43 µg/kg were recorded, respectively, in samples processed using a heat transfer material. Roasting is generally expected to reduce mycotoxin levels in peanut seeds, as reported in several studies conducted in Brazil [48], Togo [49], Senegal [50] and Nigeria [51]. In the present study, the low contamination of peanut seeds by mycotoxins did not allow for an assessment of the effect of roasting processes on the mycotoxin content of roasted peanuts.

There was no AFB1 contamination in the raw seeds used in the process without heat transfer material (<LOQ) and low contamination in those used in the process with heat transfer material (0.35 µg/kg). Despite the absence of mycotoxin contamination in raw peanut seeds used in the roasting process without heat transfer material, AFB1 were detected in the corresponding roasted peanut samples (0.50 µg/kg). Heterogeneity in seed contamination by mycotoxins, as reported by Teixido-Orries, et al. [52], could explain the variability observed in contamination levels between the raw materials used in the two roasting processes. Further studies would be valuable to assess the impact of peanut storage practices, as commonly performed in Benin, on seed contamination by mycotoxins, as well as on their distribution within a batch of seeds.

No significant difference was observed in the concentrations of AFB1 (*p*-value = 0.1825) and total aflatoxins (*p*-value = 0.1997) between roasted peanut snacks produced using roasting processes with and without heat transfer material. The use of a heat transfer material did not contribute to the mycotoxin contamination of the roasted peanuts.

It is also important to note that just-produced roasted peanut snacks do not show significantly different concentrations of mycotoxins compared to those available on the market (*p* > 0.05) (Appendix A).

### 2.3. Dietary Exposure to Aflatoxins and Risk Characterisation

#### 2.3.1. Consumer Exposure to Aflatoxins Through Marketed Peanut-Based Food Consumption

The estimated daily intakes (EDI) of AFB1 for consumers of roasted peanut snacks only, *kluiklui* only, and both products are summarised in Table 4. According to scenario 1, which was based on the median concentrations of AFB1, the median and 95th percentile daily intakes were 0.0003 and 0.001 µg/kg body weight (bw) per day, respectively, for consumers of roasted peanut snacks; 0.12 and 0.45 µg/kg bw/day for *kluiklui* consumers; and 0.09 and 0.52 µg/kg bw/day for consumers of both products. In scenario 2 (maximum AFB1 concentrations), the estimated median and 95th percentile daily intakes reached 0.04 and 0.18 µg/kg bw/day, respectively, for roasted peanut snack consumers; 0.63 and 2.45 µg/kg bw/day for *kluiklui* consumers; and 0.44 and 2.28 µg/kg bw/day for consumers of both peanut-based foods.

The risk characterisation, based on median (scenario 1) and maximum (scenario 2) concentrations of AFB1, revealed median MOE values of 1577 and 9 for roasted peanut snack consumers, 3 and 1 for *kluiklui* consumers, and 4 and 0.9 for consumers of both peanut-based foods, respectively. According to Scenario 1, 94% of roasted peanut snack consumers presented MOE values below the threshold of 10,000, while under Scenario 2, 100% of them fell below this benchmark. For all consumers of *kluiklui* and both peanut-based foods, MOE values were below 10,000 in both scenarios. Such low MOE values indicate a health concern (development of HCC) related to exposure to AFB1 among the assessed consumer groups.

In Nigeria, Oyedele, et al. [54] reported mean MOE values of 1665 for AFB1 in adult consumers of peanuts from local markets indicate a high risk of liver cancer. Kortei, et al. [55] carried out a risk assessment on peanuts and peanut-based foods collected from local markets in Ghana. The estimated daily intakes ranged from 0.068 to 0.300 µg/kg bw/day, while the corresponding MOE values varied between 1333 and 5882.

For quantitative cancer risk assessment, the estimated median liver cancer risks associated with dietary exposure to AFB1 were 1 and 166 cases per 100,000 persons per year for consumers of roasted peanut snacks, under Scenario 1 and 2, respectively (Table 4). For consumers of *kluiklui*, the estimated liver cancer risks reached 450 and 2447 cases/100,000 persons/year, and 359 and 1712 cases/100,000 persons/year for consumers of both peanut-based foods. When expressed as a proportion of the overall annual liver cancer incidence, the median risk attributed to AFB1 exposure corresponded to 0.0001% and 0.012% (Scenario 1 and 2, respectively) for roasted peanut snack consumers, 0.03% and 0.18% for *kluiklui* consumers, and 0.03% and 0.12% for consumers of both peanut-based foods.

Aydemir Atasever, et al. [56] estimated the exposure of the Turkish population to aflatoxin B1 through a daily consumption of 0.8 g of peanut paste, containing an average AFB1 concentration of 3 µg/kg. The resulting EDI was 0.03 ng/kg bw/day, corresponding to an MOE of 12,304 and an estimated annual incidence of hepatocellular carcinoma of 0.000715 cases per 100,000 persons. In Thailand, Kooprasertying, et al. [57] reported an estimated liver cancer risk potency of 0.01 cases per 100,000 individuals per year from a mean consumption of roasted peanuts of 1.3 g/day. Ezekiel, et al. [58] reported a high risk of liver cancer, estimating up to 41 cases per 100,000 population per year in Nigeria due to consumption of peanuts contaminated with AFB1.

The assessment of dietary exposure to total aflatoxins through consumption of marketed *kluiklui* and roasted peanut snacks is presented in Table 4. Based on scenarios 1 and 2, median total aflatoxins EDIs were 0.0003 and 0.04 µg/kg bw/day for roasted peanut snack consumers; 0.1 and 0.8 µg/kg bw/day for *kluiklui* consumers, and 0.1 and 0.7 µg/kg bw/day for consumers of both products, respectively.

MOE values were 1160 and 9 for roasted peanut snack consumers; 3 and 1 for *kluiklui* consumers; and 4 and 1 for consumers of both peanut-based foods, respectively, under scenarios 1 and 2. These MOE values are below the critical threshold of 10,000, indicating a real health concern. A mean MOE of 908 for AFtot was reported for Nigerian consumers of peanuts purchased from local markets [54]. Kortei et al. [55] reported EDI values ranging from 0.087 to 0.380 µg/kg bw/day and corresponding MOE values from 1053 to 4598, based on the consumption of peanuts and peanut-based foods collected from local markets in Ghana.

These estimates highlight the significant public health concern posed by aflatoxin contamination in peanut-based foods, particularly *kluiklui* marketed in Benin.

#### 2.3.2. Consumer Exposure to Aflatoxins Through “Just-Produced” Peanut-Based Foods Consumption

Since no significant difference was observed in aflatoxin concentrations between roasted peanut samples collected from markets and those just-produced, the risk assessment was re-evaluated using just-produced *kluiklui* samples from both processing methods.

The EDI of AFB1 among consumers of just-produced *kluiklui*, obtained from both processing methods, is presented in Table 5. According to scenario 1 (median concentration), the estimated median and 95th percentile of daily AFB1 intakes were 0.01 and 0.03 µg/kg bw/day, respectively, for consumers of *kluiklui* produced without maize flour; and 0.001 and 0.004 µg/kg bw/day for those consuming *kluiklui* produced with maize flour. Under scenario 2 (maximum concentration), the estimated median and 95th percentile intakes were 0.03 and 0.1 µg/kg bw/day, respectively, for consumers of *kluiklui* without maize flour; and 0.01 and 0.04 µg/kg bw/day for consumers of *kluiklui* with maize flour. The estimated median and 95th percentile intakes of AFB1 were higher through the consumption of *kluiklui* produced without maize flour.

Based on MOE approach, the median MOE values were 46 and 13 for consumers of *kluiklui* made with maize flour, and 415 and 39 for those consuming *kluiklui* without maize flour, respectively, under scenarios 1 and 2. As mentioned above, values below 10,000 are indicative of a health concern related to exposure to AFB1. The estimated liver cancer risks associated with median dietary exposure to AFB1 under scenarios 1 and 2 were 34 and 122 cases/100,000 persons/year, respectively, for consumers of *kluiklui* produced with maize flour, and 4 and 40 cases/100,000 persons/year for those consuming *kluiklui* produced without maize flour. Consumption of just-produced *kluiklui*, whether or not produced with maize flour, resulted in lower consumer exposure to AFB1 compared to *kluiklui* purchased from local markets.

Dietary exposure to AFtot through consumption of just-produced *kluiklui* is also presented in Table 5.

Based on scenarios 1 and 2, median estimated daily intakes of AFtot were 0.01 and 0.04 µg/kg bw/day for consumers of *kluiklui* made without maize flour; and 0.001 and 0.01 µg/kg bw/day for *kluiklui* made with maize flour consumers. The corresponding MOE values were 35 and 10 for *kluiklui* produced without maize flour consumers; 280 and 32 for *kluiklui* produced with maize flour consumers. MOE values below the critical threshold of 10,000 suggest a potential health concern related to dietary exposure to AFtot.

It is important to note that MOE values for AFB1 and AFtot were significantly higher for consumers of just-produced *kluiklui* compared to those of marketed *kluiklui* (*p* < 0.05), indicating a lower level of exposure to AFB1 and AFtot for consumers of just-produced *kluiklui*.

### 2.4. Dietary Exposure to OTA and Risk Characterisation

#### 2.4.1. Consumer Exposure to OTA Through Marketed Peanut-Based Foods Consumption

The daily OTA intakes of consumers of *kluiklui*, roasted peanut snacks, and both products are presented in Table 6.

Based on the median (scenario 1) and maximum (scenario 2) concentrations of OTA, the median daily intakes were 0.0001 and 1.29 µg/kg bw per day, respectively, for consumers of roasted peanut snacks; 0.01 and 0.20 µg/kg bw per day for consumers of *kluiklui*; and 0.01 and 1.52 µg/kg bw per day for consumers of both products.

Considering the risk of neoplastic effects (kidney tumours), the MOE values at the 50th percentile were 142,958 and 11 for roasted peanut snack consumers, 2115 and 74 for *kluiklui* consumers, and 2654 and 15 for consumers of both products, under scenario 1 and scenario 2, respectively. MOE values below the critical threshold of 10,000 highlight a potential health concern. Regarding non-neoplastic effects (kidney lesions), median MOE values were 46,634 and 4 for roasted peanut snack consumers, 690 and 24 for *kluiklui* consumers, and 866 and 5 for those consuming both products, under scenario 1 and scenario 2, respectively. MOE values below the threshold of 200 indicate a potential health concern.

Kouadio [60] reported an EDI of OTA ranging from 0.2 to 58 ng/kg bw/day among the Ivorian population, with corresponding MOE values for neoplastic effects varying between 72,500 and 250, based on a daily intake of 22 g of peanuts and peanut paste. Nuhu, et al. [61] assessed consumer exposure to OTA resulting from the consumption of peanut-based foods in Ghana. The EDI ranged from 0.05 to 0.25 ng/kg bw/day, with corresponding MOE values varying between 71 and 326. These low MOE values reflect a concerning public health situation in West Africa regarding chronic OTA exposure from peanut-based foods.

It is important to note, however, that only three roasted peanut samples out of 25 showed OTA concentrations above the limit of quantification. This may have led to an overestimation of consumer exposure to roasted peanuts in the two scenarios considered.

#### 2.4.2. Consumer Exposure to OTA Through “Just-Produced” Peanut-Based Foods Consumption

The median EDI of OTA for consumers of *kluiklui* without maize flour were 0.002 and 0.007 µg/kg bw/day under scenarios 1 and 2, respectively (Table 7). For consumers of *kluiklui* with maize flour, the corresponding values were 0.002 and 0.006 µg/kg bw/day. Regarding neoplastic effects (kidney tumours), MOE values derived from the median daily intake were 8832 and 2022 for consumers of *kluiklui* without maize flour, and 6639 and 2410 for those consuming *kluiklui* with maize flour, under scenario 1 and scenario 2, respectively. For non-neoplastic effects (kidney lesions), MOE values were 2881 and 660 for consumers of *kluiklui* without maize flour, and 2166 and 786 for those consuming *kluiklui* with maize flour.

Considering that 44% of *kluiklui* samples produced with maize flour (4/9) were below the limit of quantification, the exposure values estimated under both scenarios could potentially have been overestimated.

### 2.5. Microorganisms Isolated from Peanut-Based Foods

#### 2.5.1. Distribution of Moulds in Samples from Kluiklui Processing

The results of the fungal analysis conducted on samples collected during *kluiklui* production follow-up are presented in Table 8.

The mean fungal counts were 1.95, 1.42 and 1.10 log_10_ CFU/g for raw peanuts, peanut paste, and *kluiklui* produced without maize flour, respectively. For the production process involving maize flour, the corresponding fungal counts were 2.22, 1.44 and 0.83 log_10_ CFU/g for raw peanuts, peanut paste, and *kluiklui*, respectively. The maize flour used in this process showed a fungal count of 2.12 log_10_ CFU/g.

Fungal contamination was found to be more pronounced in raw peanuts than in *kluiklui*, regardless of the processing method. A similar observation was made by Norlia, et al. [62], who reported higher levels of fungal contamination in raw peanuts (0.3–3.6 log CFU/g) compared to peanut-based products (0.6–2.7 log CFU/g). In contrast, Hussain, et al. [63] reported higher fungal counts in peanut-based products than in raw peanuts in Pakistan, with mean fungal loads of 4.41 log_10_ CFU/g (2.6 × 10^4^ CFU/g) in raw peanuts and 4.57 log_10_ CFU/g (3.7 × 10^4^ CFU/g) in peanut cake.

In a previous study conducted in Benin, Adjou et al. [28] reported mean fungal counts ranging from 1.0 to 8.1 × 10^2^ CFU/g corresponding to 2.0 to 2.9 log_10_ CFU/g, in *kluiklui* samples. In the present study, the *kluiklui* samples produced using both processing methods exhibited lower total fungal counts than those reported by Adjou et al. [28].

Samples collected from the production process incorporating maize flour showed higher total mould counts and a greater fungal diversity, particularly in raw peanuts and maize flour, with 73 and 140 fungal isolates identified, respectively. The number of *Aspergillus* isolates was also higher in the peanut paste from this processing method (49 isolates), compared to the corresponding product from the process without maize flour (Table 8).

No link is observed between the prevalence of *Aspergillus* spp. in samples from *kluiklui* processing and the detected and quantified mycotoxins in this study. In raw peanuts used in the process without maize flour, no mycotoxins were detected and quantified, despite the total number of *Aspergillus* spp. isolates (32 isolates). On the contrary, in *kluiklui* samples from the process with maize flour, although only one *Aspergillus* spp. was isolated, AFB1, AFB2, AFG1, FUM B1, FUM B2, OTA, and OTB were detected and quantified.

#### 2.5.2. Fungal Species Isolated on Samples from Kluiklui Processing

Several fungal species were isolated from samples from *kluiklui* processing (Figure 2). These included *Aspergillus chevalieri*, *A. flavus*, *Aspergillus niger*, *Aspergillus* spp., *Botryodiplodia theobromae*, *Circinella muscae*, *Cladosporium* spp., *Rhizopus* spp., and *Talaromyces tumuli*. In raw peanut samples, *A. chevalieri* was the most dominant species, followed by *A. niger*. For samples from the process without maize flour, *A. flavus* predominated in both peanut paste and *kluiklui*; *T. tumuli* was the second most common species in peanut paste, whereas *A. niger* was the second most prevalent in *kluiklui*. In contrast, peanut paste and *kluiklui* obtained from the process incorporating maize flour were more heavily contaminated with *A. chevalieri*, while the maize flour itself was predominantly contaminated with *A. chevalieri* and *Aspergillus* spp.

In a previous study conducted in Benin, Adjou et al. [28] identified *A. flavus*, *A. parasiticus* and various *Aspergillus* spp., *Fusarium* spp., and *Penicillium* spp. as the dominant fungal species isolated from *kluiklui*. Similarly, Hussain et al. [63] reported that *A. flavus* was the most frequently isolated fungal species in peanut-based products, accounting for 29.2% of isolates, followed by *A. niger* (21.1%), *Aspergillus. fumigatus* (16.5%), and *Penicillium notatum* (1.6%).

Although the presence of *A. chevalieri* is not frequently reported in studies related to fungal contamination of peanuts and their derived products, some investigations have nevertheless documented its occurrence and mechanisms of action in these products. Kamarudin and Zakaria [64], after isolating and identifying fungi from marketed peanuts in Malaysia, reported that 50% of the isolates were similar to *A. chevalieri*. The authors concluded that the presence of *A. chevalieri* on peanuts creates favourable conditions for the growth of less xerophilic *Aspergillus* species, as well as other spoilage-related fungal genera, particularly mycotoxin-producing species, thereby potentially increasing the risk of mycotoxin contamination. Indeed, xerophilic fungi produce metabolic water, which favours an increase in water activity, thereby promoting the growth of other species [65]. *Aspergillus chevalieri* is a ubiquitous soilborne fungus, considered as one of the most xerophilic and xerotolerant fungal species, and is frequently associated with spoilage of nuts, dried peanuts, dried beans, spices, and stored cereals and grains [66,67,68]. *A. chevalieri* has been reported to produce gliotoxin and sterigmatocystin [69]. This may account for its high prevalence in peanuts and peanut-derived products observed in the present study. However, despite its widespread occurrence in the analysed samples, only very low concentrations of sterigmatocystin were detected.

Results of the ammonia vapour test carried out on samples from *kluiklui* processing, regardless of the process method (Appendix A), revealed a high proportion of toxigenic *A. flavus* isolates, particularly in peanut paste (31 isolates) and *kluiklui* (9 isolates). This high prevalence of toxigenic strains may explain the significant contamination observed in peanut paste and *kluiklui* from both processing methods.

#### 2.5.3. Distribution of Moulds and Fungal Isolates on Samples from Roasted Peanut Snack Processing

Table 9 shows the concentration of moulds in the roasted peanut samples. In raw peanuts, fungal counts were 2.09 and 1.85 log_10_ CFU/g for samples used in the roasting process without and with heat transfer material, respectively, with *T. tumuli*, *A. niger*, and *Rhizopus* spp. identified as the predominant fungi.

Roasted peanut snacks produced without heat transfer material showed a fungal count of 0.80 log_10_ CFU/g, with no fungal genera or isolates detected. In contrast, snacks roasted with heat transfer material had a fungal count of 1.30 log_10_ CFU/g, with four fungal genera identified (*T. tumuli*, *A. niger*, *B. theobromae*, and *Penicillium citrinum*), and a total of four fungal isolates recovered.

Although roasting without heat transfer material appears to be more effective in eliminating fungal contaminants (Table 9).

When comparing mycotoxin levels in samples from roasted peanut snack processing, it seems that there is no link between mould and the prevalence of *Aspergillus* spp. and quantified mycotoxins. Although *T. tumuli*, *A. niger*, and *Rhizopus* spp. were isolated in the raw seeds used in the process without heat transfer material, no mycotoxins were detected in these samples. And, in the raw seeds used in the process with heat transfer material, despite the presence of *Fusarium* spp., only AFB1 was detected.

## 3. Conclusions

The present study revealed a high level of mycotoxin contamination in peanut-based foods marketed in Southern Benin. AFB1, AFB2, AFG1, AFG2, as well as OTA and OTB, were the predominant mycotoxins detected in roasted peanut snacks and *kluiklui*. Mycotoxin contamination was found to be higher in *kluiklui* samples than in roasted peanut snacks, in marketed and just-produced samples. *kluiklui* collected from local markets were found to be the most contaminated, compared to those collected immediately after production. The storage conditions of *kluiklui* appear to be a critical factor contributing to its contamination with mycotoxins. During the production of *kluiklui*, the grinding of roasted peanuts in public grinders significantly contributed to its contamination with mycotoxins, especially when the grinding was carried out in grinders exclusively dedicated to peanut processing. Similarly, fungal contamination was more pronounced in *kluiklui*, with significantly higher populations of aflatoxigenic fungi, predominantly *A. flavus*, found in *kluiklui* samples produced without maize flour; whereas *A. chevalieri*, *Cladosporium* spp., and *Talaromyces tumuli* were the main species identified in *kluiklui* obtained from the processing method incorporating maize flour. The results of this study also indicate a high level of exposure and significant health concern related to aflatoxins and OTA intake through the consumption of *kluiklui*, particularly those marketed. In contrast, exposure levels among consumers of just-produced *kluiklui* and roasted peanuts were significantly lower, meaning that mycotoxin production occurs during storage after processing. Estimated liver cancer risks are considered as high, especially for frequent consumers and individuals with chronic hepatitis B infection. In the current context of climate change and global warming, favourable to fungi proliferation, it is essential to improve the storage conditions of *kluiklui* as well as the hygienic quality of the grinding equipment used during its production, in order to limit fungal and mycotoxin contamination. Such improvements are crucial to reduce consumer health risks, particularly the development of HCC, as well as kidney tumours and lesions.

## 4. Methods

### 4.1. Study Area

This study was conducted in southern Benin. Benin is located at 6°28′ N and 2°36′ E in West Africa. Benin has a tropical monsoon climate in the south and a tropical wet and dry climate in the north, according to the Koppen-Geiger climate type [41]. Benin is one of the most vulnerable countries in global warming scenarios of +2.7 °C by 2070 [42]. Climate change remains the main driving factor in the production of mycotoxins [70,71,72].

### 4.2. Sampling of Peanut-Based Foods from Beninese Markets

In order to determine mycotoxin concentrations in peanut-based foods marketed in Southern Benin, a total of 52 samples of peanut-based foods were purchased randomly in municipalities representative of peanut production and/or consumption in southern Benin (Abomey, Aplahoué, Covè, Cotonou, Glazoué, and Ouèssè). Figure 3 shows the sampling areas. The collected samples represent the main peanut-based foods produced and consumed in Southern Benin, and include roasted peanut (*n* = 27) and *kluiklui* (pressed fried peanut cake) (*n* = 25). The sampling was performed at production and retail sites (market and supermarket). The samples were ground, packed in sealed containers and stored under dry conditions for further analysis.

### 4.3. Follow-Up of Kluiklui and Roasted Peanut Snacks Production and Sampling

#### 4.3.1. Experimental Design

Figure 4 and Figure 5 illustrate the process for *kluiklui* and roasted peanut snack production, respectively. *Kluiklui* can be produced by adding or not adding maize flour during the kneading of the peanut paste. About 10 to 20 g of maize flour per kg of peanut paste is used. Regarding roasted peanuts, roasting can be done using a heat transfer material or not.

The comparison between both processes for *kluiklui* production is particularly relevant, not so much because of the potential role of maize flour as a source of mycotoxins, but rather due to the type of grinder employed. In Covè, the production process without maize flour relies on a grinder dedicated exclusively to peanut grinding, whereas in Aplahoué, the process incorporating maize flour involves the use of a mixed-use grinder. To assess the contribution of peanut processing practices to mould and mycotoxin contamination, *kluiklui* and roasted peanut snacks production follow-up was conducted on 2 producing sites (Covè and Aplahoué). The study considered both processing methods for *kluiklui* (with or without the addition of maize flour) and for roasted peanuts (with or without the use of a heat transfer material). Three independent productions, involving three different processors, were carried out for each product and processing method, resulting in a total of 36 trials (Table 10). The 12 processors were selected based on their experience in processing peanuts into *kluiklui* or roasted peanut snacks.

#### 4.3.2. Samples Collected During Follow-Up Experiments

The raw peanuts (unshelled dried seeds) were purchased from a single peanut producer based in Covè. The crop was harvested in November 2023 and underwent two post-harvest drying phases: a first sun-drying in the field for 7 days, followed by a second drying at the producer’s residence for 14 days. After drying, the seeds were stored unshelled in polypropylene bags.

As mentioned before, *kluiklui* and roasted peanut snack production were performed 3 times by each processor, at intervals of two months between the first and second repetition, and one month between the second and third repetition.

Figure 6 illustrates the distribution of raw material between the 12 selected processors, for one trial. Before each trial, shelling was carried out by the peanut producer using a mechanical sheller. Shelling was stopped once 300 kg of shelled peanuts had been obtained. The shelled seeds were then packaged in 25 kg bags before being distributed to the processors. The 25 kg of shelled seeds were sun-dried and roughly sorted (by visual examination to remove impurities and bad seeds) by each processor before being used for *kluiklui* or roasted peanut snack production.

For mycotoxin analysis, raw peanuts (shelled, dried and sorted seeds) as well as roasted peanuts, peanut paste, maize flour, and *kluiklui* were sampled in the follow-up of *kluiklui* production, while raw peanuts and samples of the final product were collected in the follow-up of roasted peanut snack production.

As the raw material was of the same origin, samples of raw peanuts (shelled, dried and sorted seeds) were pooled. For laboratory analyses, 12 pooled samples of this raw material were created by mixing the 3 samples taken from each trial. Pooling was performed in the laboratory. After freeze-drying and grinding of each individual sample, pools were formed by combining three 50 g subsamples using a blender to obtain a homogeneous mixture. Appendix A presents details of the pooling of raw materials. When pooled raw material samples tested positive for mycotoxins, the corresponding individual samples were analysed, and the resulting data were considered in the present study (Appendix A).

During the production of *kluiklui*, roasted peanut samples were collected immediately after partial roasting and cooling, and just before the dehulling and grinding of seeds. Peanut paste was obtained by grinding the roasted seeds using public grinding services. Samples were collected immediately after grinding. The maize flour used in the process was sourced from household stocks and was produced by grinding maize grains purchased from local markets. It was sampled as well. *Kluiklui* samples, produced by frying the shaped and pressed peanut paste, were collected immediately after frying and cooling.

Roasted peanut snacks were obtained by precooking and roasting peanut seeds either with or without a heat transfer material (white clay). Samples of roasted peanut snacks were collected immediately after roasting and cooling of seeds.

### 4.4. Determination of Mycotoxins in Peanut-Based Foods Samples

#### 4.4.1. Standards and Chemicals

Analytical grade standards and solvents were used in this study and sourced from various suppliers. The solvents and reagents included: acetonitrile AR (Biosolve, Dieuze, France), mass spectrometry grade acetonitrile (Biosolve, Dieuze, France), acetic acid AR (Fisher Chemical, Merelbeke, Belgium), MS/MS grade methanol (Biosolve, Dieuze, France), formic acid (Avantor, VWR, Leuven, Belgium), ammonium acetate (Fluka, Mint Street, NC, USA), magnesium sulphate (MgSO_4_) and sodium chloride (NaCl), both from Roth (Karlsruhe, Germany). Ultrapure water was obtained using a Milli-Q^®^ IQ 7010 purification system (MilliporeSigma, Burlington, MA, USA). Certified analytical standards of mycotoxins were obtained from Romer-Labs (Getzersdorf, Austria) for the following compounds: 15-acetyldeoxynivalenol (15Ac-DON), 3-acetyldeoxynivalenol (3Ac-DON), deoxynivalenol (DON), HT-2 toxin, alternariol (AOH), alternariol monomethyl ether (AME), T-2 toxin, zearalenone (ZEA), aflatoxins B1, B2, G1 and G2, ochratoxin A (OTA), ochratoxin B (OTB), fumonisins B1 (FUM B1), B2 (FUM B2) and B3 (FUM B3), as well as sterigmatocystin (STE). Standards for enniatins A, A1, B, and B1 were purchased from Merck (Merck KGaA, Darmstadt, Germany).

#### 4.4.2. Determination of Mycotoxins

Mycotoxin concentration in marketed *kluiklui* and roasted peanut snacks and in samples from production follow-up were determined using liquid chromatography tandem mass spectrometry (LC-MS/MS), according to the method developed by Jonard, et al. [73]. A QuEChERS extraction procedure was used for extraction by adding 10 mL of ultrapure water to 5 g of ground sample, accurately weighed into 50 mL polypropylene centrifuge tubes. Then, 10 mL of acetonitrile containing 1% (*v*/*v*) acetic acid was added, and samples were homogenised using a Mixer Mill MM400 (Retsch GmbH, Aartselaar, Belgium) at 10 Hz for 10 min. Subsequently, 4 g of anhydrous magnesium sulphate (MgSO_4_) and 1 g of sodium chloride (NaCl) were added. The tubes were manually shaken and agitated using the Mixer Mill MM400 for 2 min at 10 Hz. The mixtures were then centrifuged at 3000× *g* for 5 min using a refrigerated centrifuge (Centrifuge 5810 R, Eppendorf, Leipzig, Germany). A 100 μL aliquot of the resulting supernatant was collected and diluted with 900 μL of a 50:50 (*v*/*v*) methanol–water solution. The final extracts were transferred into chromatographic vials for analysis. Liquid chromatography–tandem mass spectrometry (LC-MS/MS) analyses were carried out using a Waters ACQUITY I-Class UPLC system (Waters, Milford, MA, USA) equipped with a BEH C18 analytical column (2.1 × 50 mm, 1.7 μm particle size; Waters), coupled to a Xevo TQ-xS mass spectrometer (Waters, Milford, MA, USA). Column temperature was maintained at 50 °C. The injection volume was 2 µL, and the flow rate was set at 0.3 mL/min. An elution gradient was applied using mobile phase A consisting of an ammonium acetate-water solution containing 0.1% of formic acid, and mobile phase B composed of methanol with 0.1% of formic acid. The gradient started at 10% B, held for 3 min and increased linearly to 70% over 7 min. Then, the gradient increased again from 10% in 10 s and remained constant for 2 min. Finally, B decreased to 10% in 10 s followed by an equilibration period of 3 min.

Multiple reaction monitoring mode under positive ionisation mode was used on the mass spectrometer. The capillary voltage was set to 3.4 kV, with a desolvation temperature of 400 °C. The desolvation gas flow and cone gas flow were maintained at 800 L/H and 150 L/H, respectively, and the nebuliser pressure was 7 bar.

The software TargetLynx V4.2 (Waters Corporation, Milford, MA, USA) was used to perform chromatographic peak integration and quantification.

#### 4.4.3. Method Validation

Average recoveries for each mycotoxin were evaluated for peanut, peanut paste and maize flour by spiking “blank” samples containing no mycotoxin above the limit of detection (LOD). A total of 6 “blank” samples were extracted and analysed on three different days. Reproducibility and repeatability were thus also evaluated. LOD and recovery in the different matrices are presented in Table 11. Intra-day and inter-day variability are presented in Table 12.

### 4.5. Fungal Analysis

#### 4.5.1. Fungal Enumeration and Isolation

Fungi were enumerated and isolated from samples collected during the follow-up of *kluiklui* and roasted peanut snack production. For each sample, 10 g were suspended in 90 mL of Buffered Peptone Water (BPW, Bio-Rad, Hemel, Belgium) and homogenised at 230 rpm for 2 min using a stomacher to obtain a 1:10 dilution. Serial dilutions (10^−2^ and 10^−3^) were then prepared by transferring 1 mL from the previous dilution into 9 mL of sterile BPW, following the protocol described in ISO 6887-1:2017 [74]. Moulds were enumerated on Yeast Extract Glucose Chloramphenicol (YGC) agar (Bio-Rad), incubated at 25 °C for 5 days, according to ISO 21527-2:2008 [75].

The samples were isolated directly on Water Agar (Micro Agar, Duchefa Biochemie, Haarlem, The Netherlands). Samples were also directly plated on Water Agar (Micro Agar, Duchefa Biochemie, The Netherlands). The fungal colonies observed were transferred to Potato Dextrose Agar (PDA, Scharlau, Barcelona, Spain) and sub-cultured repeatedly until pure isolates were obtained. The plates were incubated at 22 °C for 7 days. The colonies were examined under a microscope for a preliminary identification based on the observation of the macroscopic and microscopic characteristics of the isolates prior to molecular analyses.

#### 4.5.2. Molecular Identification

Genomic DNA was extracted from fungal mycelium using a modified cetyltrimethylammonium bromide (CTAB) protocol as described by Karthikeyan, et al. [76], and amplified by PCR using primers ITS1 and ITS4. The PCR mix was then sequenced for complete identification of the isolated strain. The obtained sequence was compared to those available in the NCBI GenBank database using BLAST algorithm.

#### 4.5.3. Ammonia Vapour Test

A qualitative method was used to identify aflatoxigenic strains of *A. flavus*. The isolates were cultivated in darkness on coconut agar medium supplemented with 0.3% methyl-β-cyclodextrin and incubated at 28 °C for 7 days. Following incubation, the cultures were exposed to ammonia vapour for 5 to 10 min. A colour change from pink to red was indicative of the strain’s ability to produce aflatoxins [77,78].

### 4.6. Estimation of Mycotoxins Daily Intake and Risk Characterisation

Exposure assessment and risk characterisation were conducted for aflatoxins B1 and total aflatoxins (B1 + B2 + G1 + G2) and for ochratoxin A, the main mycotoxins found in analysed samples.

#### 4.6.1. Estimation of Mycotoxins Daily Intake

A food consumption survey was performed to estimate the daily intake of *kluiklui* and roasted peanut snacks. The survey was conducted among 400 adult consumers in municipalities of Abomey, Aplahoué, Covè, Cotonou, Glazoué, and Ouèssè. Individual body weights of consumers and daily intake levels of *kluiklui* and roasted peanut snacks were recorded. Data collection was carried out through face-to-face interviews conducted in French and local languages, including Fon, Mahi, Adja, Nago, and Idatcha. Prior to each interview, participants were provided with a concise explanation of the study’s objectives to ensure informed participation. Daily consumption of *kluiklui* and roasted peanut snack data used in this study ranged respectively from 0.4 to 346 g, and from 0.6 to 284 g, with respective medians of 14.4 g and 9.5 g.

The estimated daily intake (EDI) of each mycotoxin was calculated based on the contamination levels determined in *kluiklui* and roasted peanut snacks, and on these consumption data collected from a previous food consumption survey [79]. For contamination levels, two scenarios were considered: scenario 1 was based on median concentrations (calculated as upper bound values) and scenario 2, considered as a worst-case scenario, was based on the maximum concentration of the mycotoxin of interest.$$EDI=\frac{Contamination\;level*\;Consumption\;rate}{Body\;weight\;(kg/person)}$$

Estimated daily intake: Estimated amount of mycotoxin ingested daily (µg/kg bw/day); Contamination level: median or maximum mycotoxin content (µg/kg); Consumption rate: amount peanut-based product ingested daily (g/day).

#### 4.6.2. Risk Characterisation Related to Mycotoxin Ingestion Through Kluiklui and Roasted Peanut Snack Consumption

##### Risk Characterisation Related to the Exposure to Aflatoxin B1 and Total Aflatoxins

Two different approaches previously established by international regulatory agencies were used to assess the potential health risks associated with the consumption of *kluiklui* and roasted peanut snacks: the Margin of Exposure (MOE) approach recommended by the European Food Safety Authority [12] and a quantitative liver cancer risk approach developed by the FAO and the WHO [53].

The MOE was calculated for AFB1 and total aflatoxins (sum of AFB1, AFB2, AFG1 and AFG2) using a Benchmark Dose (BMDL10) of 0.4 µg/kg body weight (bw)/day derived from the incidence of hepatocellular carcinomas (HCC) in male rats following exposure to AFB1 and total aflatoxins applying equal potency factors for AFB1, AFB2, AFG1 and AFG2, as recommended by EFSA [12].

The following equation was used:$$Margin\;of\;exposure=\;\frac{{BMDL}_{10}}{EDI}$$

An MOE above 10,000 is considered indicative of a low public health concern related to AFB1 exposure.

The quantitative estimation of liver cancer risk resulting from AFB1 exposure was performed by multiplying the carcinogenic potency (Pcancer) (number of cancers per year per 100,000 individuals per ng AFB1 per kg bw per day) by the estimated daily intake of AFB1 (ng AFB1 per kg bw per day). For estimating the risk, in the present study, it was assumed that 9.9% of the Beninese population is chronically affected by hepatitis B [80], which is a known cofactor enhancing the carcinogenic effect of AFB1. The following equations were used:$$Pcancer=\left({PHBsAg}^{+}*{pop.HBsAg}^{+}\right)+({PHBsAg}^{-}\;*\;{pop.HBsAg}^{-})$$

With:

PHBsAg+ = 0.3 cancers/year/100,000 individuals/ng AFB1/kg bw/day [42]; PHBsAg- = 0.01 cancers/year/100,000 individuals/ng AFB1/kg bw/day [42]; pop.HBsAg+ = fraction of the Beninese population with Hepatitis B (0.099) [43]; pop.HBsAg- = fraction of the Beninese population without Hepatitis B (0.901) [43]Cancer Risk=Pcancer∗Estimated Daily Intake

##### Risk Characterization Related to Ochratoxin a Exposure Through Peanut-Based Food Consumption

For ochratoxin A, a BMDL10 of 14.5 µg/kg bw/day was used to characterise neoplastic effects (kidney tumours in rats), whereas a BMDL10 of 4.73 µg/kg bw/day was applied to assess neurotoxic effects (kidney lesions in pigs) [59]. Margins of exposure (MOEs) below 10,000 for neoplastic effects and below 200 for neurotoxic effects are considered indicative of a health concern.

### 4.7. Data Analysis

Microsoft Excel 2019 was used to perform descriptive statistical analyses. Statistical analyses were done using R version 4.4.2 software. Mycotoxin concentration data were expressed on a wet weight basis. Upper-bound (UB) contamination data was used to carry out all calculations. Data were initially tested for normality using the Shapiro–Wilk test and for homogeneity of variances using Levene’s test, prior to conducting further statistical analyses. Differences between the two groups were analysed using an independent sample *t*-test or a Wilcoxon nonparametric test. All differences were considered statistically significant at *p* < 0.05.

## Figures and Tables

**Figure 1 toxins-17-00532-f001:**
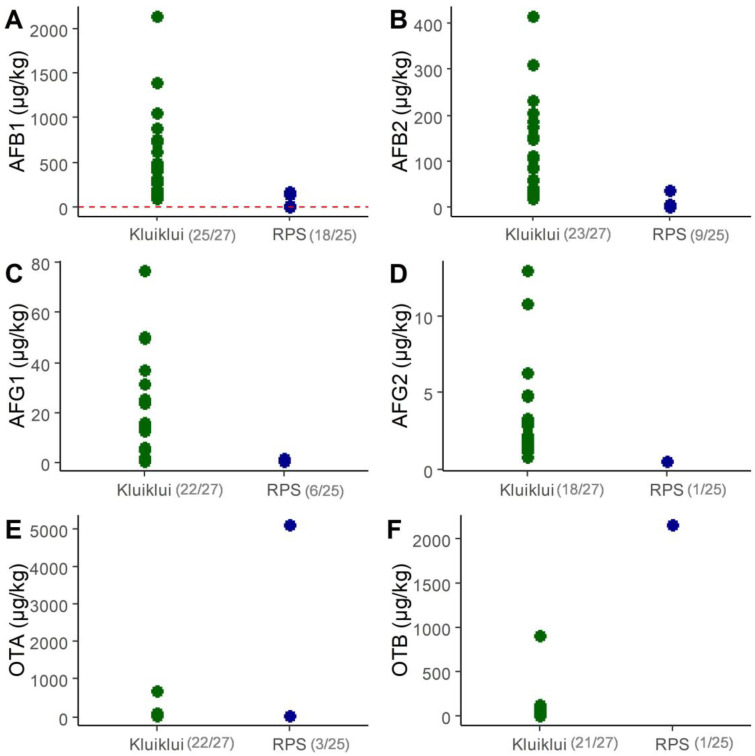
Concentrations of AFB1 (**A**), AFB2 (**B**), AFG1 (**C**), AFG2 (**D**), OTA (**E**) and OTB (**F**) in samples of marketed *kluiklui* and roasted peanut snacks (RPS). Only samples above LOQ are shown. LOQ values are 0.2 µg/kg for AFB1, AFB2, AFG1, AFG2, and 0.5 µg/kg for OTA and OTB. The total number of samples was 27 for *kluiklui* and 25 for roasted peanut snacks. The number of samples above the LOQ among the total number of samples is mentioned in brackets. In Graph A, the red dotted line represents the AFB1 limit of 2 µg/kg for peanut-based foods, as established by the European Commission (EC) [36].

**Figure 2 toxins-17-00532-f002:**
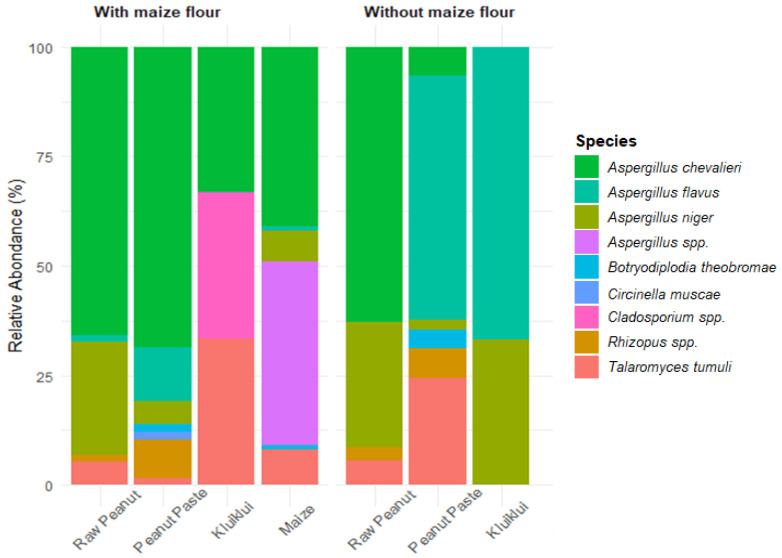
Fungal diversity associated with the samples from *kluiklui* processing.

**Figure 3 toxins-17-00532-f003:**
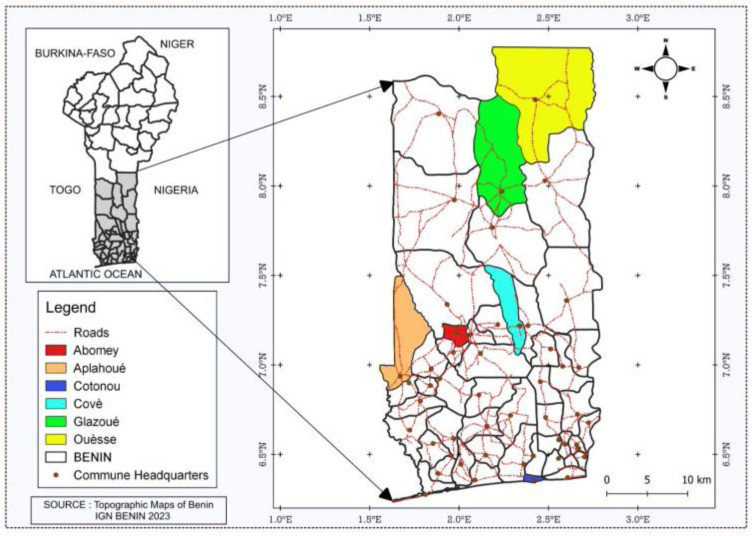
Map of Benin showing the 6 communes of the sampling areas (adapted from a map from the IGN “Institut Géographique National” of Benin using the open source QGIS software, version 3.2).

**Figure 4 toxins-17-00532-f004:**
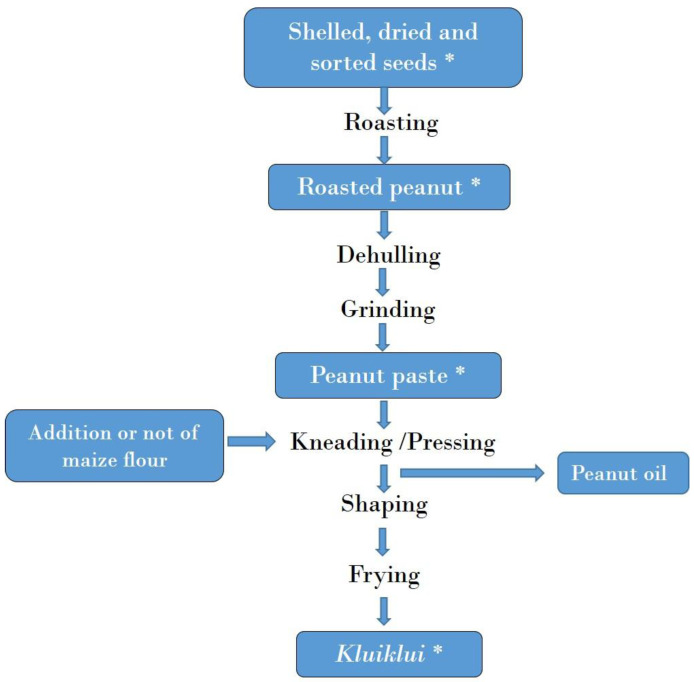
Flow chart of *kluiklui* production. *: Samples collected for mycotoxin analysis.

**Figure 5 toxins-17-00532-f005:**
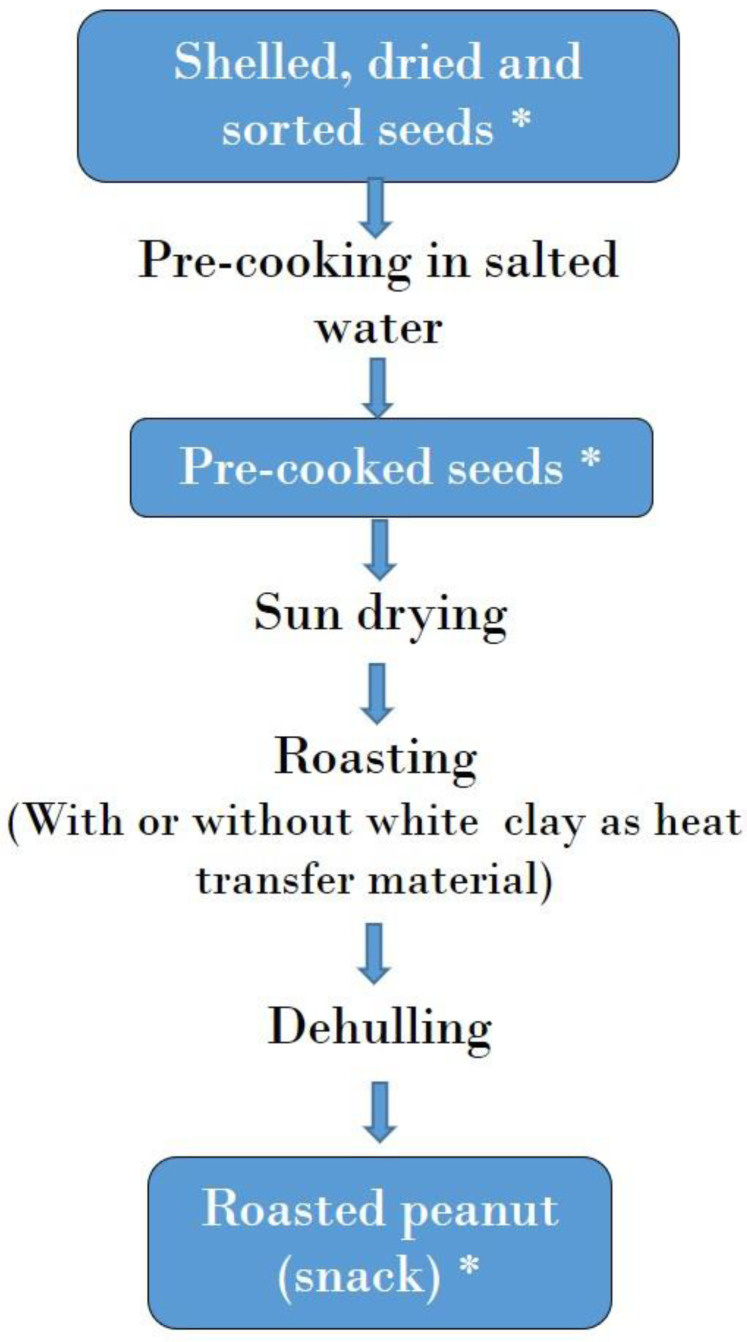
Flow chart of roasted peanut production. *: Samples collected for mycotoxin analysis.

**Figure 6 toxins-17-00532-f006:**
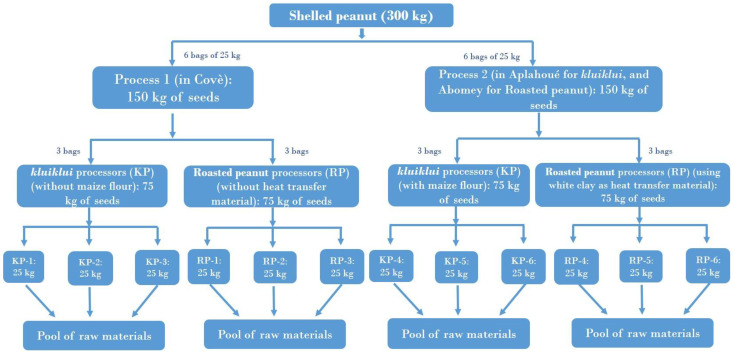
Distribution plan of raw material (shelled peanut) between the 12 processors, for each trial. This trial was performed 3 times.

**Table 1 toxins-17-00532-t001:** Mycotoxin concentrations of raw peanuts used for *kluiklui* and roasted peanut snacks (*n* = 18) (µg/kg, UB).

Parameters	AFB1 *	AFB2	AFG1	AFG2	AFtot	FUM B1	FUM B2	OTA	OTB
Sample > LOQ **	3	1	0	0	-	0	0	0	0
Min	0.02	0.07	<LOD	<LOD	0.09	<LOD	<LOD	<LOD	<LOD
Med	0.02	0.07	<LOD	<LOD	0.09	<LOD	<LOD	<LOD	<LOD
Max	2.63	0.34	<LOD	<LOD	2.97	<LOD	<LOD	<LOQ	<LOQ
% Sample > EU ML ***	0	NA	NA	NA	0	NA	NA	NA	NA

* AFB1: Aflatoxin B1, AFB2: Aflatoxin B2, AFG1: Aflatoxin G1, AFG2: Aflatoxin G2, AFtot: Total aflatoxin (AFB1 + AFB2 + AFG1 + AFG2), FUM B1: Fumonisin B1, FUM B2: Fumonisin B2, OTA: Ochratoxin A, OTB: Ochratoxin B; ** Sample > LOQ: Number of samples above limit of quantification (AFB1, AFB2, AFG1, AFG2: 0.2 µg/kg; FUM B1, FUM B2: 10 µg/kg; OTA, OTB: 0.5 µg/kg); LOD were 0.02, 0.07, 0.16 and 0.13 µg/kg for AFB1, AFB2, AFG1 and AFG2, respectively; and 0.4 µg/kg for OTA and OTB. The upper bound (UB) concentrations, where left-censored values were replaced by the LOQ or LOD, are presented for AFB1 and AFB2. *** EU ML: Maximum limits of 8.0 and 15.0 µg/kg for AFB1 and Total Aflatoxin for raw peanuts, which are not directly consumed, but subjected to a processing [36]; N/A, not applicable: No maximum limits (ML) existed for this mycotoxin for peanut-based foods.

**Table 2 toxins-17-00532-t002:** Mycotoxin contamination (µg/kg, UB) in samples collected during *kluiklui* processing.

Process	Sample	AFB1 *	AFB2	AFG1	AFG2	AFtot	AOH	FUM B1	FUM B2	OTA	OTB
Without maize flour (Covè)	Roasted peanut (*n* = 9)	<LOD (0)	<LOD (0)	<LOD (0)	<LOD (0)	-	<LOD (0)	<LOD (0)	<LOD (0)	<LOD (0)	<LOD (0)
	Peanut paste (n = 9)	50.56 ± 19.74 ^a^ (9)	7.82 ± 2.29 ^a^ (9)	2.20 ± 1.00 ^a^ (6)	0.33 ± 0.30 (1)	60.91 ± 21.98 ^a^ -	10.31 ± 4.00 (1)	<LOD (0)	<LOD (0)	3.70 ± 0.11 ^a^ (7)	1.79 ± 0.13 ^a^ (6)
	*Kluiklui* (*n* = 9)	37.78 ± 14.35 ^1^ (9)	7.37 ± 2.26 ^1^ (9)	1.60 ± 0.20 ^1^ (5)	0.23 ± 0.18 (1)	46.99 ± 16.88 ^1^ -	10.20 ± 3.80 (1)	4.09 ± 2.93 (0)	<LOQ (0)	7.05 ± 3.71 ^1^ (7)	3.37 ± 1.97 ^1^ (7)
With maize flour (Aplahoué)	Roasted peanut (n = 9)	<LOD (0)	<LOD (0)	<LOD (0)	<LOD (0)	-	<LOD (0)	<LOD (0)	<LOD (0)	<LOD (0)	< LOD (0)
	Peanut paste (*n* = 9)	14.65 ± 9.54 ^b^ (9)	6.58 ± 6.77 ^a^ (9)	0.25 ± 0.15 ^b^ (1)	<LOD (0)	21.60 ± 10.08 ^b^ -	<LOD (0)	<LOD (0)	<LOD (0)	6.13 ± 5.00 ^a^ (4)	2.94 ± 3.99 ^a^ (4)
	Maize flour (n = 6)	24.11 ± 17.08 (6)	2.61 ± 1.97 (6)	0.42 ± 0.01 (4)	<LOD (0)	27.26 ± 19.06 -	<LOD (0)	991.51 ± 277.26 (6)	333.84 ± 78.72 (6)	2.32 ± 3.71 (2)	0.62 ± 0.46 (2)
	*Kluiklui* (*n* = 9)	11.73 ± 7.09 ^2^ (9)	4.14 ± 2.29 ^1^ (9)	0.24 ± 0.14 ^2^ (1)	<LOD (0)	16.24 ± 7.87 ^2^ -	<LOD (0)	31.28 ± 47.84 ^1^ (3)	14.14 ± 9.50 ^1^ (2)	6.70 ± 2.71 ^1^ (5)	4.51 ± 6.09 ^1^ (5)

* AFB1: Aflatoxin B1, AFB2: Aflatoxin B2, AFG1: Aflatoxin G1, AFG2: Aflatoxin G2, AFtot: Total aflatoxin (AFB1 + AFB2 + AFG1 + AFG2), AOH: Alternariol, FUM B1: Fumonisin B1, FUM B2: Fumonisin B2, OTA: Ochratoxin A, OTB: Ochratoxin B Upper bound values (UB): concentrations below the limit of quantification (LOQ) or the limit of detection (LOD) were replaced by the LOQ or LOD value. LOQ were 0.2 µg/kg for AFB1, AFB2, AFG1 and AFG2; 10 µg/kg for AOH, FUM B1 and FUM B2; and 0.5 µg/kg for OTA and OTB. LOD were 0.02, 0.07, 0.16, 0.13, 8, 2.4 and 8 µg/kg for AFB1, AFB2, AFG1, AFG2, AOH, FUM B1 and FUM B2, and 0.4 µg/kg for OTA and OTB, respectively. The number of samples above the LOQ is mentioned in brackets. Values within the same column followed by the same letter or number do not differ significantly at the 5% probability level.

**Table 3 toxins-17-00532-t003:** Mycotoxin concentration in roasted peanut snacks according to the process (µg/kg, UB).

Roasting Process	Sample	AFB1 *	AFB2	AFG1	AFG2	AF_tot_	OTA
Without heat transfer material	Raw peanut (*n* = 9)	<LOQ (0)	<LOD (0)	<LOD (0)	<LOD (0)	-	<LOD (0)
	Roasted peanut snack (n = 9)	0.50 ± 0.29 ^a^ (3)	0.20 ± 0.09 ^a^ (2)	<LOD (0)	<LOD (0)	1.56 ± 1.14 ^a^ -	<LOD (0)
With heat transfer material	Raw peanut (*n* = 9)	0.35 ± 0.47 (1)	<LOQ (1)	<LOD (0)	<LOD (0)	0.37 ± 0.53 -	<LOQ (0)
	Roasted peanut snack (*n* = 9)	0.12 ± 0.12 ^a^ (1)	0.11 ± 0.04 ^a^ (2)	<LOD (0)	<LOD (0)	0.43 ± 0.51 ^a^ -	<LOD (0)

* AFB1: Aflatoxin B1, AFB2: Aflatoxin B2, AFG1: Aflatoxin G1, AFG2: Aflatoxin G2, AFtot: Total aflatoxin (AFB1+ AFB2+ AFG1+ AFG2), OTA: Ochratoxin A, OTB: Ochratoxin B. Upper-bound values (UB): concentrations below the limit of quantification (LOQ) or the limit of detection (LOD) were replaced by the LOQ or LOD value. LOQ of 0.2 µg/kg for AFB1, AFB2, AFG1 and AFG2; 10 µg/kg for FUM B1 and FUM B2; and 0.5 µg/kg for OTA and OTB. LOD of 0.02, 0.07, 0.16, 0.13 and 0.4 µg/kg for AFB1, AFB2, AFG1, AFG2 and OTA, respectively. The number of samples above the LOQ is mentioned in brackets. Values within the same column followed by the same letter do not differ significantly at the 5% probability level. The mean concentrations of AFB1 (*p* = 0.1825) and total aflatoxins (*p* = 0.1997) were not significantly different in roasted peanut snacks produced using roasting processes with and without heat transfer material.

**Table 4 toxins-17-00532-t004:** Dietary exposure of marketed peanut-based foods consumers to aflatoxin B1 and total aflatoxins.

Consumer Category	Descriptive Level	Aflatoxin B1						Total Aflatoxins			
		Estimated Daily Intake (EDI) (µg/kg bw/day)		MOE BMDL_10_ * = 0.4 µg/kg bw per Day		Cancer Risk ** (Cases/100,000 persons/year)		Estimated Daily Intake (EDI) (µg/kg bw/day)		MOE BMDL_10_ * = 0.4 µg/kg bw per Day	
		Scenario 1	Scenario 2	Scenario 1	Scenario 2	Scenario 1	Scenario 2	Scenario 1	Scenario 2	Scenario 1	Scenario 2
Consumers of roasted peanut snacks only											
*n* = 50	Min	0.00002	0.004	16,327	97	0.1	16	0.00003	0.004	10,205	96
	P50	0.0003	0.04	1577	9	1	166	0.0003	0.04	1160	9
	P95	0.001	0.18	385	2	4	681	0.001	0.18	283	2
	Max	0.003	0.49	139	0.8	11	1887	0.004	0.49	102	0.8
Consumers of *kluiklui* only											
*n* = 117	Min	0.01	0.05	47	9	33	178	0.01	0.06	39	7
	P50	0.12	0.63	3	1	450	2447	0.14	0.78	3	1
	P95	0.45	2.45	1	0.2	1748	9495	0.55	3.02	1	0.1
	Max	1.55	8.42	0.3	0.05	6002	32,604	1.88	10.38	0.2	0.04
Consumers of roasted peanut snacks and *kluiklui*											
*n* = 233	Min	0.003	0.03	135	13	11	119	0.004	0.02	111	18
	P50	0.09	0.44	4	0.9	359	1712	0.11	0.66	4	1
	P95	0.52	2.28	0.8	0.2	1997	8811	0.63	3.51	1	0.1
	Max	1.25	5.77	0.3	0.1	4845	22,326	1.52	8.73	0.3	0.05

* EFSA [12]: MOE above 10,000 means a low health concern. ** Cancer Risk: Risk of HCC incidence per year, resulting from dietary AFB1 intake through peanut-based foods consumption [53], Scenario 1: Calculation of dietary exposure based on the median concentrations (UB) of AFB1 and total aflatoxins in roasted peanut snacks (1.0 and 1.36 µg/kg) and *kluiklui* (394.82 and 478.36 µg/kg). Scenario 2: Calculation of dietary exposure based on the maximum concentrations of AFB1 and total aflatoxins in roasted peanut snacks (169.0 and 169.36 µg/kg) and *kluiklui* (2144.64 and 2644.14 µg/kg).

**Table 5 toxins-17-00532-t005:** Dietary exposure of “*just produced*” *kluiklui* consumers to aflatoxin B1 and total aflatoxins.

Products	Descriptive Level	Aflatoxin B1						Total Aflatoxins			
		Estimated Daily Intake (EDI) (µg/kg bw/day)		MOE BMDL_10_ * = 0.4 µg/kg bw per Day		Cancer Risk ** (Cases/100,000 persons/year)		Estimated Daily Intake (EDI) (µg/kg bw/day)		MOE BMDL_10_ * = 0.4 µg/kg bw per Day	
		Scenario 1	Scenario 2	Scenario 1	Scenario 2	Scenario 1	Scenario 2	Scenario 1	Scenario 2	Scenario 1	Scenario 2
*kluiklui* made without maize flour											
	Min	0.001	0.002	630	175	2	9	0.001	0.003	477	144
	P50	0.01	0.03	46	13	34	122	0.01	0.04	35	10
	P95	0.03	0.1	12	3	131	472	0.04	0.1	9	3
	Max	0.1	0.4	3	1	450	1620	0.2	0.5	2.6	0.8
*kluiklui* made with maize flour											
	Min	0.0001	0.001	5698	532	0.3	3	0.0001	0.001	3842	437
	P50	0.001	0.01	415	39	4	40	0.001	0.01	280	32
	P95	0.004	0.04	107	10	14	155	0.01	0.05	72	8
	Max	0.01	0.1	31	3	50	532	0.02	0.2	21	2

* EFSA [12]: MOE above 10,000 means a low health concern, for neoplastic and non-neoplastic effects, respectively. ** Cancer Risk: Risk of HCC incidence per year, resulting from dietary AFB1 intake through kluiklui consumption [53]. Scenario 1: Calculation of dietary exposure based on the median concentrations (UB) of AFB1 and total aflatoxins in *kluiklui* without maize flour (29.57 and 39.04 µg/kg) and *kluiklui* with maize flour (3.27 and 4.85 µg/kg). Scenario 2: Calculation of dietary exposure based on the maximum concentrations of AFB1 and total aflatoxins in *kluiklui* without maize flour (106.53 and 129.37 µg/kg) and *kluiklui* with maize flour (35.0 and 42.63 µg/kg).

**Table 6 toxins-17-00532-t006:** Dietary exposure to OTA through marketed peanut-based food consumption and MOE.

Consumer Category	Descriptive Level	Dietary Exposure to OTA (µg/kg bw/day)		MOE (Neoplastic Effects) BMDL_10_ * = 14.5 µg/kg bw/day		MOE (Non-Neoplastic Effects) BMDL_10_ * = 4.73 µg/kg bw/day	
		Scenario 1 *	Scenario 2 *	Scenario 1	Scenario 2	Scenario 1	Scenario 2
Consumer of roasted peanut snacks only							
*n* = 50	Min	0.00001	0.12	1,479,613	116	482,660	38
	P50	0.0001	1.29	142,958	11	46,634	4
	P95	0.0004	5.29	34,847	3	11,367	1
	Max	0.001	14.67	12,564	1	4099	0.3
Consumer of *kluiklui* only							
*n* = 117	Min	0.0005	0.01	29,039	1011	9473	330
	P50	0.01	0.20	2115	74	690	24
	P95	0.03	0.76	545	19	178	6
	Max	0.10	2.62	159	6	52	2
Consumer of both roasted peanut snacks and *kluiklui*							
*n* = 233	Min	0.0002	0.1	80,894	234	26,388	76
	P50	0.01	1.52	2654	15	866	5
	P95	0.03	5.25	475	3	155	1
	Max	0.10	12.83	195	1	64	0.4

* EFSA [59]: reference BMDL. For neoplastic effects, an MOE above 10,000 or above 200 means a low health concern, for neoplastic and non-neoplastic effects, respectively. Scenario 1: Calculation of dietary exposure based on the median concentrations (UB) of OTA in roasted peanut snacks (0.40 µg/kg) and *kluiklui* (23.26 µg/kg). Scenario 2: Calculation of dietary exposure based on the maximum concentrations of OTA in roasted peanut snacks (5085.0 µg/kg) and *kluiklui* (668.0 µg/kg).

**Table 7 toxins-17-00532-t007:** Dietary exposure to OTA through “*just-produced*” peanut-based food consumption and MOE.

Consumer Category	Descriptive Level	Dietary Exposure to OTA (µg/kg bw/day)		MOE (Neoplastic Effects) BMDL_10_ * = 14.5 µg/kg bw/day		MOE (Non-Neoplastic Effects) BMDL_10_ * = 4.73 µg/kg bw/day	
		Scenario 1	Scenario 2	Scenario 1	Scenario 2	Scenario 1	Scenario 2
Consumer of *kluiklui* without maize flour							
*n* = 117	Min	0.0001	0.0005	121,267	27,762	39,558	9056
	P50	0.002	0.007	8832	2022	2881	660
	P95	0.006	0.03	2276	521	743	170
	Max	0.02	0.1	663	152	216	50
Consumer of *kluiklui* with maize flour							
*n* = 117	Min	0.0002	0.0004	91,155	33,094	29,735	10,796
	P50	0.002	0.006	6639	2410	2166	786
	P95	0.008	0.02	1711	621	558	203
	Max	0.03	0.10	498	181	163	59

* EFSA [59]. For neoplastic effects, an MOE above 10,000 or above 200 means a low health concern, for neoplastic and non-neoplastic effects, respectively. Scenario 1: Calculation of dietary exposure based on the median concentrations (UB) of OTA in *kluiklui* without maize flour (5.57 µg/kg) and *kluiklui* with maize flour (7.41 µg/kg). Scenario 2: Calculation of dietary exposure based on the maximum concentrations of OTA in *kluiklui* without maize flour (24.33 µg/kg) and *kluiklui* with maize flour (20.41 µg/kg).

**Table 8 toxins-17-00532-t008:** Mould and prevalence of *Aspergillus* spp. in samples from *kluiklui* processing.

Process	Sample	Mold Count (Log_10_ CFU/g) *	Total Number of Genera *	Total Number of Fungal Isolates	Total Number of *Aspergillus* spp. Isolates	Detected and Quantified Mycotoxins
Without maize flour	Raw peanut (*n* = 9)	1.95 ± 0.00	3	35	32	-
	Peanut paste (*n* = 9)	1.42 ± 0.33	4	45	29	AFB1, AFB2, AFG1, AFG2, AOH, OTA, OTB
	*Kluiklui* (*n* = 9)	1.10 ± 0.41	2	15	15	AFB1, AFB2, AFG1, AFG2, AOH, OTA, OTB
With maize flour	Raw peanut (*n* = 9)	2.22 ± 0.00	3	73	6	-
	Peanut paste (*n* = 9)	1.44 ± 0.19	5	57	49	AFB1, AFB2, AFG1, OTA, OTB
	Maize flour (*n* = 6)	2.12 ± 0.31	2	140	136	AFB1, AFB2, AFG1, FUM B1, FUM B2, OTA, OTB
	*Kluiklui* (*n* = 9)	0.83 ± 0.06	1	3	1	AFB1, AFB2, AFG1, FUM B1, FUM B2, OTA, OTB

* Log_10_ CFU/g: Logarithm of colony forming units (CFU) per gram of sample; Total number of genera: Total number of fungal genera isolated; Total number of fungal isolated: Total number of fungal strains isolated; Total number of *Aspergillus* spp. isolated: Total number of *Aspergillus* strains isolated.

**Table 9 toxins-17-00532-t009:** Mould count and concentration of AFtot in samples from roasted peanut snack processing.

Roasting Process	Samples	Mould Count (Log10 CFU/g)	Total Number of Genera	Total Number of Fungal Isolates	Fungal Isolates	Detected and Quantified Mycotoxins
Without heat transfer material	Raw peanut (*n* = 3)	2.09 ± 1.04	3	7	*T. tumuli* *A. niger*, *Rhizopus* spp.	-
	Roasted peanut snack (*n* = 3)	0.80 ± 0.17	0	0	-	AFB1, AFB2
With heat transfer material	Raw peanut (*n* = 3)	1.85 ± 0.14	3	5	*A. niger* *Rhizopus* spp. *Fusarium* spp.	AFB1
	Roasted peanut snack (*n* = 3)	1.30 ± 0.52	4	4	*T. tumuli* *A. niger*, *B. theobromae* *P. citrinum*	AFB1, AFB2

**Table 10 toxins-17-00532-t010:** Experimental design applied to the production of *kluiklui* and roasted peanuts.

Peanut-Based Foods	Municipality	Processing Technology Considered	Number of Processors	Number of Trials
*Kluiklui*	Covè	Technology excluding maize flour during kneading of the peanut paste	3	9 *
	Aplahoué	Technology involving maize flour addition during kneading of the peanut paste	3	9
Roasted peanut snack	Covè	Roasting without heat transfer material	3	9
	Abomey	Roasting with heat transfer material	3	9

* 3 processors per processing technology and 3 trials per processor.

**Table 11 toxins-17-00532-t011:** Limit of detection (LOD) and recovery of mycotoxin of interest in the different matrices.

Mycotoxin	LOD (µg/kg)	Recovery in Roasted Peanuts (%)	Recovery in Peanut Paste (%)	Recovery in Maize Flour (%)
Aflatoxin B1	0.024	74	69	95
Aflatoxin B2	0.066	69	67	88
Aflatoxin G1	0.16	68	63	94
Aflatoxin G2	0.13	69	67	99
Fumonisin B1	2.4	58	51	100
Fumonisin B2	8.0	67	60	69
Fumonisin B3	6.8	64	56	110
Ochratoxin A	0.5	84	91	120
Ochratoxin B	0.5	82	88	107

**Table 12 toxins-17-00532-t012:** Average intra-day repeatability and inter-day reproducibility of mycotoxin of interest in roasted peanuts.

Mycotoxin	Variation Coefficient of Intra-Day Repeatability (%)	Variation Coefficient of Inter-Day Reproducibility (%)
Aflatoxin B1	5.9	5.0
Aflatoxin B2	5.6	7.9
Aflatoxin G1	4.8	5.1
Aflatoxin G2	6.9	8.6
Fumonisin B1	17.0	25.1
Fumonisin B2	7.8	13.1
Fumonisin B3	2.5	3.7
Ochratoxin A	7.8	10.4
Ochratoxin B	3.1	7.3

## Data Availability

The original contributions presented in this study are included in the article/Appendix A. Further inquiries can be directed to the corresponding author(s).

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
