# Peer review of "Peanut and Peanut-Based Foods Contamination by Toxigenic Fungi and Mycotoxins: Potential Risks for Beninese Consumers"

_toxins, 2025, doi:10.3390/toxins17110532_

Round 1

Reviewer 1 Report

Comments and Suggestions for Authors

The manuscript ID: toxins-3931370 "Peanut and Peanut-Based Foods Contamination by Toxigenic Fins: Potential Risks for Beninese Consumers" presents fungal and mycotoxin contamination in raw peanuts harvested in November 2023 and thermally processed (roasted peanut snacks and Kluiklui) in Benin, Africa. Contamination was determined by microbiological, mycotoxicological, and molecular methods. The highest contamination was detected in thermally processed products, and it was considered that the uncleaned public grinder is the source of cross-contamination between samples. The highest health risk (cancer) was determined for Kluiklui consumers. The conclusions highlight the need to clean the public grinder and improve the storage conditions of Kluiklui on the sales market. 

The results are interesting and useful, and I suggest some improvements to the manuscript. Please find the review comments in the attached document.

All the best.

Reviewer 2 Report

Comments and Suggestions for Authors

Dear editor,

The work is well structured and the bibliographic references are consistent with the content of the text. Therefore, this reviewer suggests that this study is potentially suitable for publication in Toxins. The manuscript is acceptable for publication if some minor revisions are made to improve the quality of the content and writing. Some comments on the content are:

Line 6. Clearly explain what kluiklui is in both the abstract and the introduction.

Line 38. The discussion on Penicillium-derived mycotoxins would benefit from the inclusion of additional relevant reference, particularly the following study: Garello et al., Several secondary metabolite gene clusters in the genomes of ten Penicillium spp. raise the risk of multiple mycotoxin occurrence in chestnuts, Food Microbiology, Volume 122, 2024, 104532, https://doi.org/10.1016/j.fm.2024.104532

Line 79. The cited regulation should be formatted as follows: Commission Regulation (EU) 2023/915.

Lines 130, 132, 150. Figures 1, 2, and 3 are comprehensive from an experimental standpoint; however, their visual presentation could be improved. Enhancing the aesthetic quality of these figures would help increase the overall clarity and visibility of the work.

Line 192. The chapter “Determination of Mycotoxins” lacks essential information regarding the chromatographic method employed and the mass spectrometry settings used. Including these details is crucial to ensure the reproducibility of the results and a comprehensive understanding of the experimental approach.

Round 2

Reviewer 1 Report

Comments and Suggestions for Authors

The manuscript was improved.